# Psychometric Properties and Confirmatory Factor Analysis of the Spanish Version of the Maudsley Violence Questionnaire among Adolescent Students

**DOI:** 10.3390/ijerph18158225

**Published:** 2021-08-03

**Authors:** Vanesa Pérez-Martínez, Miriam Sánchez-SanSegundo, Rosario Ferrer-Cascales, Oriol Lordan, Nicola Bowes, Carmen Vives-Cases

**Affiliations:** 1Community Nursing, Preventive Medicine, Public Health and History of Science Department, University of Alicante, 03009 Alicante, Spain; vanesa.perez@ua.es (V.P.-M.); carmen.vives@ua.es (C.V.-C.); 2Health Psychology Department, University of Alicante, 03009 San Vicente del Raspeig, Spain; rosario.ferrer@ua.es; 3Business Organization Department, Universitat Politècnica de Catalunya, 08222 Terrassa, Spain; oriol.lordan@upc.edu; 4Cardiff School of Sport and Health Sciences, Cardiff Metropolitan University, Cardiff CF5 2YB, UK; nbowes@cardiffmet.ac.uk

**Keywords:** machismo, acceptance of violence, MVQ, dating violence, teenagers

## Abstract

The Maudsley Violence Questionnaire (MVQ) is an instrument specifically developed to evaluate violent thinking through two subscales examining macho attitudes and the acceptance of violence. This study analyzed the psychometric properties and factor structure of the Spanish version of the MVQ questionnaire in a large sample of 1933 Spanish adolescents. An online questionnaire was used to collect variables, such as sociodemographic and sexism data. The factor structure showed good fit indices in Spanish adolescents, which were similar to the original scale. The exploratory analysis yielded a first factor that explained 11.3% of the total variance and a second factor that explained 10.8% of the total variance. The Goodness of Fit Index (GFI) (0.902), Adjusted Goodness of Fit Index (AGFI) (0.90), Normed Fit Index (NNFI) (0.85), and the Comparative Fit Index (CFI) (0.86) suggested that the model fit the data adequately (with values ≥ 0.90) and the Root Mean Square Error of Approximation (RMSEA) (≤0.10) values indicative of an adequate fit. This study contributes a Spanish-language validated tool to measure machismo and the acceptance of violence among adolescents.

## 1. Introduction

Teen dating violence (TDV) refers to physical, psychological, and/or sexual violence occurring in the context of an adolescent romantic relationship between young people who do not cohabit or have legal or marital ties [1]. TDV includes all act of controlling behaviors, emotional blackmail, stalking, intimidation, sexual abuse, and physical aggression [2]. This type of violence has become a public health problem, with immediate and long-term health implications for adolescents who are victims of violence.

Studies examining the prevalence of dating violence among adolescents and youth adults have reported that nearly one third of victims of physical and/or sexual violence were women in an age range between 15 and 24 years (WHO, 2013) [3]. Although boys and girls report similar rates of perpetrating ‘mild’ violence, acts of severe violence are predominantly committed by males; with male perpetration almost twice that of females [4].

In Europe, and Spain in particular, the prevalence of physical and/or sexual violence is similar to the statistics reported worldwide. For example, a study carried out with a Spanish sample of 4337 students between 15 and 26 years old revealed that 26.4% of adolescents had been victims of abuse, with psychological abuse being the most prevalent form of aggression [5]. A study carried out with European adolescents showed that TDV victimization was higher in girls (34.1%) than in boys (26.7%), and there was a higher prevalence of psychological violence (control and fear) over physical and/or sexual violence, especially in girls (28.1%) compared with in boys (21%) [6].

The recognition of abuse among adolescents during a dating relationship has been demonstrated to play an important role in the escalation of violence. However, most young people experience difficulties in perceiving abuse within their relationship. It has been estimated that nearly 70% of young women do not perceive physical and psychological abuse in their dating relationship because they often tend to attribute these behaviors as an expression of affection and love [7,8,9].

The research literature to date has shown that adolescents who are targets of TDV, are at an increased risk of experiencing adverse outcomes, including fear, poor academic performance, and social rejection [10]. Likewise, adolescent victims of TDV are more likely to experience difficulties with their mental health, including anxiety, depression, suicide, and future intimate partner violence [11,12,13].

TDV has a profound impact not only at the individual level, but also at the economic and social level [14]. Studies examining the economic impact of intimate partner violence suggested that the cost associated with healthcare exceeds 19 million dollars per 100,000 women in United States [14]. In addition, the different forms of TDV may affect interpersonal relationships [14].

In the last two decades, different assessment instruments have been developed aimed at assessing early behaviors that promote violence, including aggressive behaviors [15], sexist attitudes [16], and antisocial behaviors [17]. However, there is a paucity in measures designed to analyze the role of cognition, beliefs, and violent thoughts as precursors of violence in adolescent couples. This is a problem because cognition, beliefs, and violent thoughts are all features of theoretical explanations and models of violence.

Some scales, such as the Firestone Assessment of Violent Thoughts [18] and its version for adolescents (FAVT-A), have been shown to be effective in measuring negative and hostile thoughts. However, these scales assess automatic thinking styles, and not the more conscious rules or cognitive distortions that promote violence and violent behavior [18]. Additional scales, such as the NOVACO scale, [19], the Criminal Attitudes to Violence scale [20], and the Psychological Inventory of Criminal Thinking Styles [21], have been mainly developed for adult and criminal populations, and therefore their use with adolescents is not recommended.

From this perspective, one of the most recently developed instruments showing utility in identifying thought patterns and attitudes that justify the use of violence is the Maudsley Violence Questionnaire scale [22]. The MVQ has been validated for use in general adult populations [22,23,24,25,26], adolescent populations [22], and offender populations [23,25]. The items of MVQ were generated from a comprehensive approach of cognitive behavioral formulation of violence, which suggests that violence is used as a legitimate strategy to protect low self-esteem [24].

The MVQ questionnaire is comprised of two factors measuring (i) macho attitudes; and (ii) attitudes that normalize violence [22]. Previous studies supported a significant relationship between the macho attitudes of dating violence and the perpetration of TDV [25,26]. It was found that girls and boys who display macho attitudes are at increased risk of being a victim of dating violence [27]. Violence tolerance in a relationship has been identified as a risk factor for the increased frequency of abuse and perpetration of violence [28,29].

In this study, we examined the psychometric properties of a Spanish version of the MVQ scale in a large sample of adolescent students who were part of the European Project “Lights, Camera and Action against Dating Violence” (Lights4Violence) [30] and the “Promotion of gender violence protective assets among adolescents and preadolescents” funded by the Spanish Ministry of Economy, Industry and Competitiveness and the Carlos III Institute (Ref. PI18/00590 and PI18/00544), continuation of Lights4Violence [30]. We hypothesized that the Spanish MVQ scale would show convergent validity via positive correlations with sexism and that the psychometric properties of the Spanish version would be similar to those obtained in the original MVQ English version [22].

## 2. Materials and Methods

### 2.1. Participants

The initial sample consisted of 2207 students selected from seven public high schools in eastern Spain, of which 1150 were boys (*SD* = 0.001) and 1057 were girls (*SD* = 0.001). Participants ranged in age between 13 to 18 years (*M* = 14.15; *SD* = 1.21). A statistical power analysis was carried out to estimate the sample size, with the support of previous published validation studies, where the recommendation of sample size was from 5 to 10 participants per item [31]. Participants completed an online questionnaire distributed to the schools from March 2018 to October 2019.

The inclusion criteria for participation in the study were: (i) being enrolled in a compulsory secondary education program, (ii) having a good understanding of literacy in Spanish, and (iii) being present in the classroom at the time of the evaluation. Those participants with language and/or literacy problems that could obstruct the understanding of the questionnaires or did not for allow their completion were excluded. During the evaluation period, 13 participants (0.59%) refused to take part in the study, and 211 (9.56%) did not complete the questionnaire. The 50 remaining questionnaires could not be analysed due missing data in responses for some of the variables or in certain items of the scale itself (2.26%). The final sample was composed of 1933 students.

### 2.2. Measures

Sociodemographic data were collected, including age, sex, place of birth of the students, nationality, and socioeconomic variables, including the employment situation of the parents and their educational level. Other information collected related to the exposure to violence throughout their life, whether or not participants had been in a dating relationship, experiences of abuse and/or violence before 15 years of age by an adult, and if they had been victims of dating violence.

#### 2.2.1. Maudsley Violence Questionnaire (MVQ)

The MVQ [22] was used to measure violent thinking. The MVQ is a scale designed to evaluate thoughts, cognitions (mental schemas), and distortions that justify violent attitudes and behaviours. It is comprised of 56 items separated in to two subscales: machismo (42 items) and the acceptance of violence (14 items). Responses are rated on a dichotomous scale of true (1) or false (0) in a range from 0 to 56 items. Machismo is the main predictor of violence, particularly in men—acceptance does not predict violence in men and only partially seems to be related to female violence [23].

Higher scores on the machismo subscale are indicators of violent thinking. The machismo subscale includes items that represent statements related to the male role, such as the integration of violence (e.g., “being violent shows that you are a man”), situations considered as shameful also related to the fact of fleeing from the violent situation (e.g., “it is embarrassing to leave a fight”), and statements that justify its use (e.g., “it is OK to hit someone who makes you feel stupid”).

The acceptance of violence subscale includes items related to the acceptance of violence (e.g., “It is ok watching violence on TV) and the justification of violence (e.g., “It is OK to hit someone who threatens your partner”). In terms of psychometric properties, the English original version of the instrument demonstrated good internal consistency with an alpha coefficient between good and excellent for the factors machismo and acceptance of violence (0.73 to 0.913). The reliability was obtained for men and women separately, being higher for men in the machismo factor [22].

#### 2.2.2. The Ambivalent Sexism Inventory (ASI)

This scale was developed by Glick and Fiske (1996) [16] and is composed of a 22-item self-reported measure of sexism. It is composed of two sub-scales: the hostile sexism scale, which is composed of 11 items designed to evaluate an individual’s position on the dimensions of competitive gender differentiation, dominative paternalism, and heterosexual hostility, and benevolent sexism scale, which is comprised of a further 11 items that aim to assess an individual’s position on the dimensions of protective paternalism, complementary gender differentiation, and heterosexual intimacy.

Respondents indicate their level of agreement, on a six-point Likert-type scale. The Spanish version [32] obtained good psychometric properties: the hostile sexism subscale obtained an 0.89 alpha coefficient, and the benevolent sexism subscale obtained 0.86. The reliability for the total scale was 0.90. The ASI scale had satisfactory convergent, discriminant, and predictive validity.

### 2.3. Cross-Cultural Adaption to Spanish

The process of translation and adaptation of the MVQ instrument was carried out following the principles of good practice, developed by the Translation and Cultural Adaptation group (TCA), belonging to the Quality of Life Special Interest Group (QoL-SIG) from ISPOR, owned by the University of Oxford [33]. For the translation, the services of two native Spanish translators were required, of which two translations into Spanish were obtained.

Along with these, two native English translations were also obtained. Both groups of translators rated the difficulty of translating each of the elements of the scale on a scale from 0 (low difficulty) to 5 points (high difficulty). The MVQ scale showed a mean difficulty of 2.3 (SD 0.56), ranging from 1.1 to 3.5. The differences between the items of the original scale and the items of adapted scale were subtle and mainly reflected differences at the linguistic and cultural level, and thus there was no necessary further adaptation of the translated items. The Spanish version of the measure can be found in Appendix A.

### 2.4. Interpretation Pilot Tests

To analyse the cognitive interpretation of the questionnaire, it was administered to a pilot sample of 60 students. From the participating schools, the students were randomly selected based on their gender and age among the students at the educational school to safeguard the equity between boys and girls for this phase. The pilot version of the questionnaire was administered to 30 boys (50%) and 30 girls (50%) between 13 and 15 years old. After implementing the questionnaire and performing the corresponding statistical analyses, the results yielded good reliability indices for the total scale and for both subscales (Cronbach’s alpha = 0.80–0.95). The results obtained in the pilot test suggested that there was a good understanding of the items. This version of the scale was implemented for this study because no changes were needed at the grammatical level.

### 2.5. Procedure

Information and data were collected anonymously. A unique code was created for each participant on the first date of data collection. In accordance with the Spanish “Real Decreto 1720/2007, de 21 del Reglamento de desarrollo de la Ley Orgánica 15/1999, de protección de datos de carácter personal”, informed consent was signed by school directors, parents, and students, where information was provided about the characteristics of the study and the possibility of leaving the study at any time. The online questionnaire was composed, first, of questions related to sociodemographic characteristics (i.e., sex, age, parent’s studies, and parents’ occupation) and followed by the MVQ scale. The questionnaire was individually and anonymously completed in the school during the academic course of 2019–2020.

There was no financial compensation associated with participation. If a student reported that they had experienced abuse, we followed the protocol to inform the school. The Lights4Violence protocol was approved by the ethical committee of the University of Alicante. It was also registered in ClinicalTrials.gov by the coordinator (Clinicaltrials.gov: CT03411564. Unique Protocol ID: 776905. Date registered: 18 January 2018).

### 2.6. Statistical Analyses

The data were analyzed using SPSS v.24 (IBM Corp., Armonk, New York, NY, United States) and R and the lavaan package (R foundation for Statistical Computing, Viena, Austria). The characteristics of the sample were described in terms of percentages for the nominal variables and means (SD) or medians (interquartile ranges IQRs) for the continuous variables. The skewness and kurtosis were calculated to measure the shape of the value distributions. Participants were randomly assigned to one of two groups. An exploratory factor analysis (EFA) was conducted in the half sample (n = 967) while confirmatory factor analysis (EFA) was conducted in 966 participants. The factor structure of the MVQ in the Spanish sample was conducted using maximum likelihood factoring and a Promax rotation, including all 56 items of the original scale.

Interclass correlations (ICC) were calculated in a randomized sample of 176 adolescents after 6 months to examine the temporal stability of the MVQ. For the EFA, Cronbach’s Alpha values were calculated for the total score and the scores for each subscale. The first factor of the EFA analyses showed that the MVQ scale was comprised by items describing machismo subscale, while the second factor consisted of items describing the acceptance of violence. The convergent validity was calculated using Pearson’s correlation coefficient between MVQ subscales and the scores on total sexism, benevolent sexism, and hostile sexism.

We determined the ceiling effect and floor effect indicative of the number of participants with scores higher than 95% and lower than 5%, respectively. Then, a confirmatory factor analyses (CFA) in the half sample was conducted following fit indices and criteria recommended by Meyers, Gamst, and Guarino (2013): the goodness of fit index (GFI ≥ 0.90), adjusted goodness of fit index (AGFI ≥ 0.90), the normed fit index (NFI ≥ 0.95), the incremental fit index (IFI ≥ 0.90), the comparative fit index (CFI ≥ 0.95), and the root mean square error of approximation (RMSEA ≤ 0.10). Items with factor loadings of less than 0.30 were subject to elimination. Modification indices were examined in order to improve the fit of the model.

## 3. Results

### 3.1. Descriptive Characteristics of the Sample

Once we eliminated missing data, the sample included 1933 students between 13 to 18 years old (mean: 14.07; SD: 1.233), 934 girls and 999 boys. Statistical analyses were carried out to identify differences between gender (*t*-test statistic) for each covariable. Most of the sample reported Spanish nationality, although there were missing responses (n = 22). Most of the participant’s parents had paid work, although there was a high proportion of mothers who dedicated themselves exclusively to housework (19.3%) in comparison to fathers (1.8%). Statistically significant differences were found between girls and boys in this variable (*p* = 0.036).

Regarding the educational levels of parents, most had completed secondary school or had engaged in further education or university. There was a statistically significant difference (*p* < 0.001) between the proportion of adolescents who reported knowing a female victim of intimate partner violence (38.1%) in comparison with adolescents that knew a male victim (14.2%). More than half of adolescents had been in a dating relationship (n = 1022, 52.9%), of which 51.46% were boys and 48.53% were girls. Of those adolescents who claimed to have had a relationship (n = 1022), 20.5% reported to have suffered psychological violence, where 10.7% were girls (n = 123).

Regarding being the victim of physical and/or sexual violence, 8.8% reported having been victim, where 4.9% were boys (n = 56). Differences between girls and boys were found for those who reported have been suffered physical and/or sexual dating violence (*p* = 0.022). Around a fifth of the sample reported being a victim of abuse in childhood, with 18.7% reporting to have suffered physical abuse and 4.7% reporting being a victim of sexual abuse. Differences were found between girls and boys who suffered physical abuse in childhood (*p* = 0.020). Boys were more likely to witness abuse and/or violence when compared with girls (3.7%), with significant differences (*p* = 0.003) (Table 1).

### 3.2. Mean and Standard Derivation Scores

Table 2 shows the mean and standard derivations for girls and boys in two factors (machismo and acceptance of violence) and the skewness and kurtosis for each subscale. Significant differences were found between girls and boys in both factors. Boys obtained significant higher scores than girls in machismo and acceptance of violence. Specifically, this difference in scores between girls and boys was more notable in the acceptance of violence.

### 3.3. Exploratory Factor Analysis and Reliability of the MVQ

Table 3 presents the results of psychometric analysis of the MVQ scale and its two dimensions. The results yielded a first factor that explained 11.70% of the total variance, and a second factor that explained 10.80% of the total variance. The first factor consisted of items describing the importance of violence for “being a macho” and strong, associating weakness with the non-use of violence and experiencing embarrassment over backing down (machismo subscale in the original validation study [24]). The second factor consisted of items describing the enjoyment of and acceptance of violence generally, and some violent attitudes as an acceptable behavior (the acceptance of violence subscale in the original validation study; Walker, 2005).

The two factors were moderate correlated with each other (r = 0.62), demonstrating that the machismo and acceptance of violence subscales were related but unique factors. Cronbach’s alpha values were calculated for each of the subscales and for the total scale. Referring to the machismo subscale, Cronbach’s alpha value was (*α* = 0.89); for the acceptance of violence subscale (*α* = 0.78) and for the total score (*α* = 0.91). Both subscales and the total scale demonstrated acceptable internal consistency.

We randomly subsampled 176 participants who participated after 6 months to demonstrate the temporal stability of the machismo and acceptance of violence subscales. The machismo subscale had a *α* = 0.72 (ICC = 0.71), and the target subscale had a *α* = 0.75 and temporal stability (ICC = 0.75). To examine MVQ’s convergent validity in Spanish adolescents, the MVQ subscales scores were correlated with Ambivalent Sexism Scale and its subscales. The MVQ subscales were positively associated with the total ASI scale and its subscales (hostile and benevolent sexism) in almost the full sample (n = 1422), demonstrating good convergent validity (Table 4). Two items were removed due to their loading values lower than 0.2 (item 21 “la violencia es secundaria para mí” (“violence is second nature to me”) and item 52 “No se debe pegar a alguien porque puedes provocarle sufrimiento y dolor” (“Because anyone can suffer hurt and pain, you should not hit other people”). Therefore, the final Spanish scale was composed of 54 items.

### 3.4. Structural Equation Model (CFA)

A total of 54 items of the MVQ were included in the CFA model as manifest variables. The two latent variables were the machismo and acceptance of use of violence.

The GFI (0.902), AGFI (0.90), NNFI (0.85), CFI (0.86) suggested that the model fit the data adequately (with values ≥ 0.90) and RMSEA (≤0.10) values indicative of adequate fit [34]. All items had statistically significant loadings onto their latent construct (<0.001). The correlation between the latent variables and machismo and acceptance of the use of violence was 0.66 (*p* < 0.001) suggesting that the MVQ factor structure showed adequate fit indices in Spanish adolescent students. Table 5 shows these results.

### 3.5. Invariance Testing by Gender

We examined the invariance as a function of gender (girls and boys). We compared three sets of measurement: measurement weights, measurement covariances, and measurement residuals. We found that all set of comparisons were significant (*p* < 0.01), suggesting that all three sets of measurements differed between girls and boys. The comparison for structural weights suggested that the correlation between the two latent factors was slightly higher for girls (*r* = 0.033) than for boys (*r* = 0.017). The CFA showed some differences between girls and boys, although previous studies suggested that a large sample size could contribute to magnify statistical differences [35].

## 4. Discussion

The aim of the study was to examine the psychometric properties and factor structure of the Spanish version of the MVQ scale in a large sample of adolescent students. The findings suggest that the MVQ Spanish version showed good psychometric properties, as it obtained similar results as the original validation of the scale [22]. Nevertheless, the validation of the original scale was only obtained at the exploratory level. Our study provides new data about confirmatory validation. The results showed that the Spanish version of MVQ was easily understood for the pilot participants, which did not lead to any grammatical changes after the first stage of the cross-cultural validation into Spanish.

A descriptive analysis was carried out by sex with some sociodemographic and violence variables, where the boys had been the victims of physical and/or sexual dating violence and witness of violence more than girls. According to previous research [36,37], the DV victimization is present in both sexes, with a similar prevalence. However, studies found that 20–30% of boys suffered physical aggression each year, and 70–90% suffered psychological aggression [38], while others found evidence that suggests that boys could be the victims more than girls in physical and psychological aggression [39].

In the current study, the exploratory analysis showed a factor solution of two factors reflecting statements linked to the male role (macho attitudes), and attitudes that justify violence. The first factor (machismo subscale) explained more variance (11.3%) than the second factor (acceptance of violence, 10.8%), as the original validation of the scale [22], where machismo appeared to be a collection of risk factors to violent thinking. Two items were eliminated due to their lower loading values. A possible explanation of these results could be the lack of understanding of the items, due to these items being translated with a double negative meaning.

The CFA analysis also showed a factor structure of the MVQ scale with adequate fit indices as well as good reliability and internal consistency. These findings are similar to those results obtained in the original English version carried out by Walker (2005) [22]. Additionally, our results showed a good convergent validity between the MVQ subscales of machismo and acceptance of violence subscales and the Ambivalent Sexism Inventory. These results are in line with previous studies suggesting an interaction between the acceptance of violence and benevolent sexist attitudes, which has been found to be a predictor of dating violence victimization [40].

In addition, prior studies in Spain have also found that sexism was one of the factors linked to violent behaviour within relationships [41]. The results of gender invariance in our study suggested that there were some differences between boys and girls in the CFA showing that boys reported more machismo and acceptance of violence than girls. Other studies that used the MVQ scale obtained the same results [22,27,42]. The differentiation in the socialization process between girls and boys may be an explanation of these results [43], where boys are educated to suppress emotions and display attitudes more related to violence.

The current findings provide the first results and validation of the scale with Spanish adolescents. The results indicate that the Spanish version of the MVQ could be a valuable measure, allowing researchers to identify macho attitudes and the normalization of violent behaviors among adolescents. This scale could be a useful measure to adolescents and educators due to its easy application and understanding. However, the current study has several limitations, which should be considered in future research. First, the participants of the study were recruited only from one Mediterranean area of Spain.

Therefore, it is not possible to generalize the results to the rest of the population, such as students from other countries. It could be useful to examine the psychometric properties of the MVQ scale in other Spanish and Latin American adolescents, to see if the scale obtains the same results in other Spanish-speaking adolescents. Self-report measures may be another limitation because of the social desirability phenomenon, which refers to the conscious or unconscious tendency to respond in a favorable way to others to avoid feeling judged from society [44].

Despite this, other studies obtained that self-reported violence was significantly correlated with officially recorded violence, suggesting that people were reporting accurately [23]. Future studies should consider this and examine machismo and the acceptance of violence among adolescents using an alternative method. Third, the current study did not examine the association between the MVQ scale and other violence self-reported scales. Future research should assess additional violence scales.

## 5. Conclusions

Despite the limitations stated above, this study showed that the MVQ scale is suitable for use in the Spanish adolescent population. This is an important advance for the study for violence in the school setting because the use of feasible and validated scales allows for the detection of individuals at high risk of violence and aggressiveness. School provides an environment to develop prosocial abilities and interpersonal relationships. Therefore, these results highlight the need for designing and implementing educational programs focused on machismo and the acceptance of violence among adolescent students.

## Figures and Tables

**Table 1 ijerph-18-08225-t001:** Sociodemographic characteristics of the sample and violence variables by sex.

	Total *n* (%)		Girls *n* (%)	Boys *n* (%)	*p*-Value (95%CI ^1^)
**Country**					0.229
Spain	1087	56.2	514 (40.9)	573 (45.5)	
Other country	149	7.7	66 (5.2)	83 (6.6)	
**Meet a female victim of IPV ^2^**					*p* < 0.001
Yes	737	38.1	406 (21)	331 (17.1)	
No	1196	61.9	528 (27.3)	668 (34.6)	
**Meet a male victim of IPV ^2^**					0.054
Yes	274	14.2	125 (6.5)	149 (7.7)	
No	1659	85.8	809 (41.9)	850 (44)	
**Have been in a dating relationship**					0.691
Yes	1022	52.9	496 (25.7)	526 (27.2)	
No	911	47.1	438 (22.7)	473 (24.5)	
**Victim of dating violence**					
Fear and/or control	236	20.5	123 (10.6)	113 (9.8)	0.137
Physical and/or sexual	101	8.8	45 (3.9)	56 (4.9)	0.022
**Victim of abuse in childhood**					
Physical abuse	362	18.7	185 (9.6)	177 (9.2)	0.020
Sexual abuse	90	4.7	47 (2.4)	43 (2.2)	0.132
**Witness abuse and/or violence**					0.003
Yes	124	6.4	52 (2.7)	72 (3.7)	
No	1808	93.5	882 (45.7)	926 (47.9)	

^1^ CI: Confidence Interval. ^2^ IPV: Intimate Partner Violence

**Table 2 ijerph-18-08225-t002:** The means and standard deviations of the scores of the MVQ scale factors and t-test differences between girls and boys and the skewness and kurtosis for the subscales.

Factor	Girls (*SD*)	Boys (*SD*)	*t*-Value	*p*-Value	Skewness	Kurtosis
F1. Machismo	8.36 (6.57)	9.47 (7.35)	−3.48	0.001	1.22	1.56
F2. Acceptance of violence	4.66 (3.29)	5.44 (3.27)	−5.22	*p* < 0.001	0.37	−0.65

**Table 3 ijerph-18-08225-t003:** Items corresponding to each factor.

Factor	Items
F1. Machismo	1, 3, 4, 5, 7, 8, 9, 10, 11, 13, 15, 16, 19, 20, 22, 23, 24, 25, 26, 28, 30, 31, 33, 35, 36, 39, 41, 42, 43, 44, 45, 46, 47, 48, 50, 51, 53, 54, 55, 56
F2. Acceptance of violence	2, 6, 12, 14, 17, 18, 27, 29, 32, 34, 37, 38, 40, 49

**Table 4 ijerph-18-08225-t004:** Correlation matrix examining the convergent validity.

	ASI *_Total	ASI *_Hostile	ASI *_Benevolent
MVQ_Machismo	0.40 **	0.36 **	0.33 **
MVQ_Acceptance of violence	0.27 **	0.30 **	0.17 **

* Ambivalent Sexism Inventory. ** The correlation is significant at the 0.01 level (bilateral).

**Table 5 ijerph-18-08225-t005:** Goodness of fit indices for the two factor models.

Model (2 Factor)	χ^2^(df)	χ^2^/df	CFI	SRMR	RMSEA (90%CI)
Final model (54 items)	1358	1.925	0.858	0.041	0.031 (0.028–0.043)

χ^2^ (df): Chi-square statistics (degree of freedom); CFI: comparative fit index; SRMR: standard root mean square residual; RMSEA: root mean square error of approximation; CI: confidence interval.

## Data Availability

The data presented in this study are available on request from the corresponding author. The data are not publicly available due to confidential information about the schools and participants.

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
