# Peer review of "Psychometric Properties and Confirmatory Factor Analysis of the Spanish Version of the Maudsley Violence Questionnaire among Adolescent Students"

_ijerph, 2021, doi:10.3390/ijerph18158225_

Round 1

Reviewer 1 Report

Recommendation:   Accept in present form

Dear authors,

Thank you for considering me as a reviewer for this publication in your esteemed journal. I have provided my comments as follows.

Firstly I would like to inform that I don´t have any potential conflict of interest neither any other ethical concerns with regards to the paper:

Confirmatory Factor Analysis of the Spanish version of the Maudsley Violence Questionnaire Among Adolescent Students

In this manuscript authors provide a Spanish-language validated tool to measure machismo and acceptance of violence among adolescents. The clinical interest is low-average as violence and undesirable behaviors like machismo are barely related to mental (or others) illnesses. However, the sociological importance of this question is so high that this research is worth it. A sociological (rather than clinical) orientation is -to nitpick- the only drawback I found in this great work. Otherwise, it was a pleasure to review this paper as the authors have shown an expertise and thoroughness that should be appreciated. The present their results in a clearly written and well-organized way. The information provided is comprehensive and I like the way it is shown.

When you translate questionnaire items into another language, you need to obtain verification from the language expert that the translated version carries the same meaning after doing back translation. This procedure is normally carried out during pre-test period. The items should be pre-tested to the expert in the field and also the respondents and obtain their comments; And this is what the authors did.

In general I am impressed with the efforts produced and the result achieved. As a result, I recommend that the paper can be accepted for publication without further modification.

I am going to provide some minor proposed changes but, I would insist that the paper is generally well written, structured and suitable for publication even without a second round. 

In the introduction I would elaborate better on the explanation about physical, emotional and/or sexual violence occurring in the context of an adolescent romantic relationship between young people.

In line 163 “In accordance with the Spanish Royal Decree 1720/2007 on the Protection of Personal Data in children”  I would refer to it in Spanish as Real Decreto 1720/2007, de 21 del Reglamento de desarrollo de la Ley Orgánica 15/1999, de protección de datos de carácter personal instead of in English; consult the JERPH style guide or ask the editor about it.

In line 308 you missed “in” before line with

And that is all

Kind regards

The reviewer

Author Response

Thank you very much for your time and your suggestions. We hope you see them reflected in the article.

Reviewer 2 Report

This study analyzed the psychometric properties and factor structure of the Spanish version of the Maudsley Violence Questionnaire (MVQ) in 1933 Spanish adolescents. The statistical methods are reasonable.

Some suggestions for the manuscript are listed below.

Title

  1. This study examined the psychometric properties of MVQ but not factor structure only. The title should be revised.

Introduction

  1. The authors started the introduction by teen dating violence. However, MVQ is not used for teen dating violence only. A broader view for introducing the importance of surveying violence in teens is needed.
  2. The first paragraph in Introduction jumped form teen dating violence to WHO’s statistical data for the violence victimization of young women. Revision to show the connection is needed.
  3. “…this phenomenon” in the second paragraph of Introduction section is unclear.

Methods

  1. Error? “Machismo is the main predictor of violence, particularly in men – acceptance does not predict violence in men and only partially seems to be related to female violence [20].”

Author Response

(The authors gave the same response as above.)

Reviewer 3 Report

I think your work is very good. Its importance is clearly exposed and it is very well described, detailing every step of the performed analysis, including the assets and limitations. The only aspect I think should be improved is the references: only 9 in 41 were published in the last 5 years. Authors should add more recent references.

Author Response

(The authors gave the same response as above.)

Reviewer 4 Report

This paper entitled “Confirmatory Factor Analysis of the Spanish version of the Maudsley Violence Questionnaire Among Adolescent Students” presents empirical evidence for the adaptation and validation of a relevant measure to Spanish adolescents. Overall, the paper is well-structured and achieved informative findings. Despite my general positive outlook about this work, some points can be clarified and improved. Therefore, I present my detailed comments and suggestions below.

I think that the title is ok and it summarizes quite well the main topic.

Key-words: I would suggest replacing some key-words already provided in the title (i.e., confirmatory factor analysis; adolescents) for alternatives labels (e.g. teenagers).

The abstract is well written and structured, summarizing the main points of the work.

The Introduction comprises four paragraphs and the authors presented a satisfactory review of the state of the art about teen dating violence and Maudsley Violence Questionnaire. Nonetheless, violence constitute a major public health issue, for a better adjustment to this Journal, I recommend strengthening this connection. The general aim was properly presented in the last paragraph, but I would clarify the specific goals (i.e., for instance, there is no reference to temporal reliability; therefore, please, identify all the evidences provided).

In my opinion “Materials and Methods” section is clear, although some details can be improved. For instance, on “Participants” subsection did not report the standard-deviation concerning the distribution of gender, because it is meaningless. Besides the range, please provide the mean age of the entire sample. I would rather prefer to have description of the analyzed sample in this subsection instead of “Results”; please consider change this content.  Measures were well described. I think authors should included further information about the pilot test, namely applied procedures and what changes were introduced as an aftermath of this phase.  Concerning procedures, I think authors provided the most relevant information to replicate the study.  However, some issues may be improved. For instance, how was the study presented to the participants, what was the order of the application of the measures. Moreover, for the first time authors mentioned “first date of data collection”; more information should be provided about the temporal reliability study (e.g., interval time, instructions provided to the participants).

The “Data Analysis” is clear and easily understood.

Results” section is well structured and it provides information in accordance with the general aim that was previously defined. On the subsection “Statistical analyses”, on page 4, ll.178/179, there is a incomplete sentence (i.e., “Participants were randomly assigned to one …”). Concerning temporal reliability, besides de total score, I would recommend computing Kappa statistics (due to the nominal nature of the variables) for the individual items.  On page 5, ll. 201, please replace “mean” by “M”.  Additionally, I suggest you to include this subsection on “Materials and Methods”. On the subsection “Exploratory factor analysis and reliability” some figures have 2 and other 3 decimals; please make numbers’ presentation homogeneous. I would like to know (and that you present information) if this Spanish version had the similar structure than the original one (i.e., same items on the subscales or not?). Throughout the section “Results”, please add italics to all statistical symbols (e.g., M, n, p, r, etc.).

In the “Discussion” section, authors reviewed and explained the main findings. Overall, it meets the expectations. I think it will be interesting if the authors can discuss the two items from the original scale that were excluded in this Spanish version. The main “Limitations” were well identified and discussed. Concerning “Implications”, I would suggest including some discussion about the application of this measures at academic and applied settings.

In Table 1, please delete the symbol “%” from the figures since it was already presented in the column title and please make data presentation homogeneous - i.e., on total include the percentage on (). At the same table it is not clear what figures are presented in column “t-test (95% CI)”; please clarify this information. On Table 2, please add the significance-value. On Table 4, please add legends (e.g, ASI_Total). On Table 5, please include the values corresponding to 90% CI.

Lastly, on “References” please review them according to the Journal guidelines (e.g., reference 35). Additionally, I think that more recent literature should be reviewed and added: indeed, only 6/41 references were published in the last 5 years and this represents a weakness.

Good work and I wish the efforts made to improve this paper paid off!

Author Response

(The authors gave the same response as above.)

Reviewer 5 Report

The paper is well-written. I have just two minor comments:

  1. The first letter of the word "version" in the title should be capitalized.
  2. Table 1 is somewhat difficult to understand. There are two issues. First, it is unclear if the comparison under "Country" is between countries or between gender groups. Second, one of the numbers in the column labeled as "T-test" is indicated as a P value, while the rest are represented as raw numbers. It would help to be consistent.

Author Response

(The authors gave the same response as above.)
